

# Emerging semantics to link phenotype and environment

Anne E. Thessen[1,2], Daniel E. Bunker[3], Pier Luigi Buttigieg[4],
Laurel D. Cooper[5], Wasila M. Dahdul[6], Sami Domisch[7], Nico M. Franz[8],
Pankaj Jaiswal[5], Carolyn J. Lawrence-Dill[9], Peter E. Midford[10],
Christopher J. Mungall[11], Martín J. Ramírez[12], Chelsea D. Specht[13],
Lars Vogt[14], Rutger Aldo Vos[15], Ramona L. Walls[16], Jeffrey W. White[17],
Guanyang Zhang[8], Andrew R. Deans[18], Eva Huala[19], Suzanna E. Lewis[11] and
Paula M. Mabee[6]

[1] Ronin Institute for Independent Scholarship, Monclair, NJ, United States
[2] The Data Detektiv, Waltham, MA, United States
[3] Department of Biological Sciences, New Jersey Institute of Technology, Newark, NJ, United States
[4] HGF-MPG Group for Deep Sea Ecology and Technology, Alfred-Wegener-Institut, Helmholtz-Zentrum für Polar-und Meeresforschung, Bremerhaven, Germany
[5] Department of Botany and Plant Pathology, Oregon State University, Corvallis, OR, United States
[6] Department of Biology, University of South Dakota, Vermillion, SD, United States
[7] Department of Ecology and Evolutionary Biology, Yale University, New Haven, CT, United States
[8] School of Life Sciences, Arizona State University, Tempe, AZ, United States
[9] Departments of Genetics, Development and Cell Biology and Agronomy, Iowa State University, Ames, IA, United States
[10] Richmond, VA, United States
[11] Genomics Division, Lawrence Berkeley National Laboratory, Berkeley, CA, United States
[12] Division of Arachnology, Museo Argentino de Ciencias Naturales–CONICET, Buenos Aires, Argentina
[13] Departments of Plant and Microbial Biology & Integrative Biology, University of California, Berkeley, CA, United States
[14] Institut für Evolutionsbiologie und Ökologie, Universität Bonn, Bonn, Germany
[15] Naturalis Biodiversity Center, Leiden, The Netherlands
[16] iPlant Collaborative, University of Arizona, Tucson, AZ, United States
[17] US Arid Land Agricultural Research Center, United States Department of Agriculture—ARS, Maricopa, AZ, United States
[18] Department of Entomology, Pennsylvania State University, University Park, PA, United States
[19] Phoenix Bioinformatics, Redwood City, CA, United States

Corresponding author
Anne E. Thessen,
annethessen@gmail.com

## ABSTRACT

Understanding the interplay between environmental conditions and phenotypes is a fundamental goal of biology. Unfortunately, data that include observations on phenotype and environment are highly heterogeneous and thus difficult to find and integrate. One approach that is likely to improve the status quo involves the use of ontologies to standardize and link data about phenotypes and environments. Specifying and linking data through ontologies will allow researchers to increase the scope and flexibility of large-scale analyses aided by modern computing methods. Investments in this area would advance diverse fields such as ecology, phylogenetics, and conservation biology. While several biological ontologies are well-developed, using them to link phenotypes and environments is rare because of gaps in ontological coverage and limits to interoperability among ontologies and disciplines. In this manuscript, we present (1) use cases from diverse disciplines to illustrate questions that could be answered more efficiently using a robust linkage between phenotypes and environments, (2) two proof-of-concept analyses that show the

value of linking phenotypes to environments in fishes and amphibians, and (3) two proposed example data models for linking phenotypes and environments using the extensible observation ontology (OBOE) and the Biological Collections Ontology (BCO); these provide a starting point for the development of a data model linking phenotypes and environments.

## INTRODUCTION

Phenotype is the expression of interactions between genotype and environment. This relationship is fundamental to a wide range of biological research, from large-scale questions about the effect of climate change on global ecosystems to small-scale questions involving disease presentation in individual organisms. The urgency of answering such questions, coupled with the "data deluge" (*Hey, Tansley & Tolle, 2009*), has motivated scientists to explore more efficient ways to aggregate and explore life science data. Traditional methods of data dissemination, publication, and deposition in stand-alone databases do not support the rapid, automated, and integrative methods of data exploration needed to efficiently address pressing research priorities.

Two important barriers to understanding interactions between environment and phenotype are the heterogeneity of terms and their imprecise definitions in data sets and manuscripts. An ontology has the potential to tame this heterogeneity and allow researchers to more efficiently query and manipulate large-scale data sets (Fig. 1); however, several challenges must be overcome before their benefits are realized. Historically, bio-ontologies first came into popular use in the biomedical and model organism communities (*Côté & Robboy, 1980*; *Spackman, Campbell & Côté, 1997*; *Ashburner et al., 2000*), but they are now being applied to address much broader, comparative problem complexes (*Mabee et al., 2007*; *Deans et al., 2015*; *Dececchi et al., 2015*). A shift towards representing and reasoning over taxonomically diverse phenotypes in an ontological framework has occurred in a period of less than 10 years and, not surprisingly, has brought about new semantic, computational, and even social challenges (*Gerson, 2008*). In this paper, we explore the difficulties of automated linking of environments and phenotypes, review the current state-of-the-art, present use cases, and propose solutions to frequently encountered problems.

Clearly representing the natural language descriptions of phenotypes and environments with a set of ontologies is difficult, because natural language, while highly expressive, is often semantically ambiguous and reliant on context (*Sasaki & Putz, 2009*; *Seltmann et al., 2013*). Despite successes in developing standards within specific disciplines (e.g., *Taylor et al., 2008*), standard vocabularies are rare and seldom widely or consistently used (*Enke et al., 2012*). Individual scientists often have preferred terms with undocumented and highly nuanced meanings (*Chang & Schutze, 2006*; *Huang et al., 2015*). Further, there

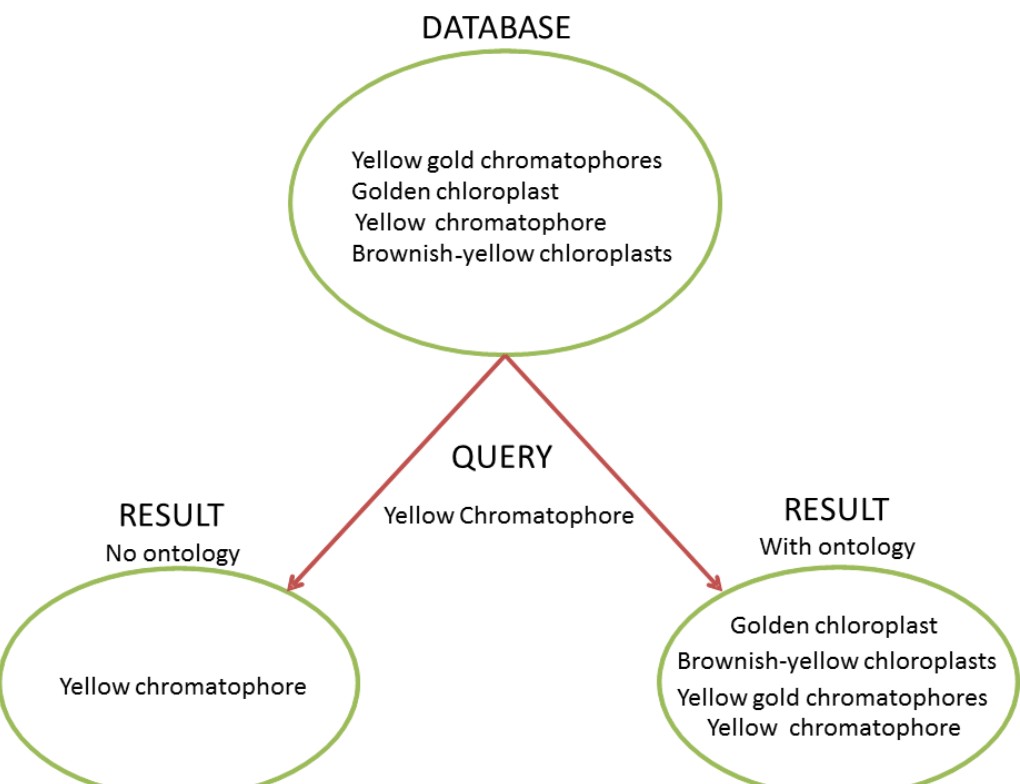

**Figure 1 Ontology and the Heterogeneity Challenge.** This diagram demonstrates how ontologies can solve the challenge of heterogeneous terminology. In this example, the database contains four different natural language descriptions about dinoflagellate chloroplasts harvested from text. A user needs to query the database for instances of dinoflagellates with yellow chromatophores. Without an ontology to provide information to the query-engine about synonymy ("chromatophore" = "chloroplast") and term relationships ("brownish-yellow", "golden", and "yellow gold" are subtypes of "yellow"), a query for "yellow chromatophore" will only yield one of the four results the user needs and would find using an ontology. Without an ontology to link closely related concepts with a common parent, and reconcile heterogeneous terms, a user would have to perform many more queries to get a desired result, which may not be tractable in a large dataset.

is a co-evolution between natural language and ontologies, which can complicate the recording of provenance and reduce backwards-compatibility (*Seppälä, Smith & Ceusters, 2014*; *Ochs et al., 2015*). Thus, as it stands, a researcher wishing to perform a meta-analysis has to manually integrate data sets, which often requires discussions with data providers to clarify meaning.

In addition to the intricacies of natural language used to describe phenotypes and environments, the ontological representation of environments requires additional considerations. Environments are often described using a collection of semantically complex and often ambiguous terms that are applied differently across disciplines. Semantic representation of terms such as "environment," "ecosystem," "habitat," "ecozone," "bioregion," and "biome" must account for variable biological, ecological, geographic, geopolitical, and historical usage. As a result of this complexity, many specialized environmentally-themed terms such as "Afrotropical," which are central to fields such

as zoogeography and floristic science, are not yet included in any ontology. Data about species interactions and behavior can be an essential component in defining an organism's environment, but current ontological structures do not include behavior regulation classes that can be tied to ecological processes (e.g., negative regulation of foraging behavior by predation pressure). Environments can also be defined using a more data-driven approach where a specific environment is defined as the intersection of different factors, e.g., defining a desert as having a specific annual precipitation, temperature range, and solar irradiation. In the field of plant science, field management practices (including, e.g., frequency irrigation or fertilizer application) are sometimes considered to be a component of environment, whereas others consider field management to differ from the environment because these changes to conditions are not a part of the natural environment. Because of differences in perspective, data are organized in different ways across multiple resources. These inconsistencies in describing environments complicate analyses that might identify associations among occurrences, phenotypes and environmental conditions.

In spite of these challenges, understanding relationships between the semantics of environment and phenotype is fundamental for data integration and scientific progress in the fields of conservation, agriculture, disease control, organismal development, and numerous others in biology. Thus, there is a need for a more developed, flexible, and interlinked ontology framework representing environments, phenotypes, and their interplay. This framework for environments and phenotypes can allow automated inferencing over large, aggregated data sets, as demonstrated for gene functions and biological processes (*Ashburner et al., 2000*). Below, we present (1) use cases from diverse disciplines to illustrate questions that could be answered more efficiently using a robust linkage between phenotypes and environments, (2) two proof-of-concept analyses that show the value of linking phenotypes to environments in fishes and amphibians, and (3) two proposed example data models for linking phenotypes and environments using the extensible observation ontology (OBOE) and the Biological Collections Ontology (BCO); these serve as a starting point for the development of a data model linking phenotypes and environments.

## BACKGROUND

Within the field of informatics, classification strategies range from flat lists of terms, to vocabularies, and ontologies. For example, a vocabulary might merely contain the terms "bone," "leg," "femur" and their definitions. An ontology would further define these as classes and with respect to their biological relationships by asserting that a "femur" is a type of "bone" and part of the "leg." Moreover, such assertions are encoded in a standardized, machine-readable form. Thus ontologies empower computers to reliably interpret and reason over these logical relationship graphs. A well-known example of the technology's potential is provided by IBM's Watson (*Gliozzo et al., 2013*). Ontologies are typically recorded in a syntax format such as the Web Ontology Language (OWL; *W3C OWL Working Group, 2012*) or the Resource Description Framework (RDF; http://www.w3.org/RDF/) that can be readily distributed and exchanged by computers, thereby facilitating knowledge integration within a scientific community. For an ontology to actually be useful

to scientists, these same scientists must mutually agree upon, develop, and nurture a shared collection of ontologies and the processes for maintaining it.

Over 40 ontologies and vocabularies have been created to describe environment and phenotype (Table 1). Some of these resources are extended and refined through incorporation of user and developer requests for new terms and cross-referencing terms to existing vocabularies. Like software development, an essential aspect of ontology development is constant evaluation through active use: describing data sets and asking key biological questions. To this end, a number of groups are making initial inroads in the use of ontologies for studying the impact of environment on phenotypes (e.g., The Encyclopedia of Life; *Pafilis et al., 2015*). The Minimum Information for any Sequence (MIxS; *Yilmaz et al., 2011*) metadata checklist, a product of the Genomic Standards Consortium (GSC; *Field et al., 2011*), does not specifically link phenotypes to environments, but does include fields for describing environments using the Environment Ontology (ENVO; www.environmentontology.org; *Buttigieg et al., 2013*) for sequences from environmental samples. Sequences from parasites or endo- or epibionts can have their environments described as host phenotype using a phenotype ontology (Phenotype Quality Ontology, *Gkoutos et al., 2004*; Human Phenotype Ontology, *Köhler et al., 2014*; or Mouse Phenotype Ontology, *Gkoutos et al., 2004*). The International Consortium for Agricultural Systems Applications (ICASA) has built an infrastructure for combining genotype, environment, and management data in agricultural analyses using a hierarchical data dictionary (*Hunt, White & Hoogenboom, 2001*; *White et al., 2013*). This infrastructure is being integrated in crop and climate modeling efforts, notably through the Agricultural Model Intercomparison and Improvement Project (AgMIP), which promotes efforts to "simulate yield response to climatic factors, abiotic factors, and genotypic variables" (http://research.agmip.org/). *Oellrich et al. (2015)* recently developed a standardized method for describing and analyzing the phenotypes associated with characterized mutant genes across species that includes environmental terms from the Plant Environment Ontology (EO). Despite this progress, the available environment and phenotype ontologies still contain major gaps in the coverage of their respective domains, and significant investment is needed before data integration and analytics can be accomplished on a large scale.

## USE CASES

To communicate the importance of investing in environment and phenotype ontologies, we present use cases drawn from several life science domains. These use cases represent the types of research questions that either cannot currently be answered or can only be answered with great difficulty. For each use case, background information is discussed before presenting a current workflow, without ontologies, and a future workflow, with ontologies. A section is devoted to explaining how the ontologies helped in the future workflow and then challenges in moving from the current to the future workflow are discussed. Most of the ontologies needed to enable the future workflow either do not exist or need to be enhanced. Thus, the future workflow and the explanation of how the ontologies could help are hypothetical.

**Table 1** List of resources (vocabularies and ontologies) relevant to annotating phenotypes and environments.

| Name | Abbreviation | URL | Reference |
|---|---|---|---|
| AGROVOC | | 1 | |
| Behavioral Ontology | NBO | 2 | |
| Biological Collections Ontology | BCO | 3 | *Walls et al., 2014a* |
| Biological Spatial Ontology | BSPO | 4 | *Dahdul et al., 2014* |
| Chemical Entities of Biological Interest | ChEBI | 5 | *Hastings et al., 2013* |
| CMECS Habitat Classification | | 6 | |
| Crop Ontology | CO | 7 | *Shrestha et al., 2010* |
| Eagle-i Resource Ontology | ERO | 8 | |
| EcoLexicon | | 9 | |
| Ecological Classifications NatureServe | | 10 | |
| Environment Ontology | ENVO | 11 | *Buttigieg et al., 2013* |
| EUNIS Habitat Classification | | 12 | |
| Experimental Factor Ontology | EFO | 13 | *Malone et al., 2010* |
| Exposure ontology | EXO | 14 | |
| Fission Yeast Phenotype Ontology | FYPO | | *Harris et al., 2013* |
| Flora Phenotype Ontology | FLOPO | 15 | *Vos et al., 2014* |
| Floristic Regions of the World | | 16 | *Takhtajan, 1986* |
| Fungal gross anatomy | FAO | 17 | |
| Gazetteer | GAZ | 18 | |
| Gene Ontology | GO | 19 | *Ashburner et al., 2000* |
| GeoNames | | 20 | |
| Getty Thesaurus of Geographic Names | | 21 | |
| Global Administrative Areas | GADM | 22 | |
| Human Phenotype Ontology | HP | | *Köhler et al., 2014* |
| Information Artifact Ontology | IAO | 23 | *Ceusters, 2012* |
| International Consortium for Agricultural Systems Applications standards | ICASA | 24 | *White et al., 2013* |
| IUCN Habitats Classification Scheme | | 25 | |
| Mammalian phenotype | MP | 26 | *Smith & Eppig, 2009* |
| Mapping European Seabed Habitats | MESH | 27 | |
| NASA GCMD keyword list for locations | | 28 | |
| Observation Ontology | OBOE | 29 | *Madin et al., 2007* |
| Ontology of Biological Attributes | OBA | 30 | |
| Ontology of Biomedical Investigation | OBI | 31 | *Brinkman et al., 2010* |
| Ontology of Microbial Phenotypes | OMP | 32 | *Giglio et al., 2009* |
| Phenotype Quality Ontology | PATO | 33 | *Gkoutos et al., 2004* |
| Plant Environment Ontology | EO | 34 | |
| Plant Ontology | PO | 35 | *Cooper et al., 2013* |
| Plant Trait Ontology | TO | 36 | *Jaiswal et al., 2002* *Arnaud et al., 2012* |
| Population and Community Ontology | PCO | 37 | |
| Relation Ontology | RO | 38 | |
| Semantic Web for Earth and Environmental Terminology | SWEET | 39 | *DiGiuseppe, Pouchard & Noy, 2014* |
| Sequence Ontology | SO | 40 | *Eilbeck et al., 2005* |

| Name | Abbreviation | URL | Reference |
|------|--------------|-----|-----------|
| Terminology of Grazing Lands and Grazing Animals | | | *Allen et al., 2011* |
| Uber Anatomy Ontology | UBERON | 41 | *Mungall et al., 2012*; *Haendel et al., 2014* |
| Worm Phenotype | WBPhenotype | 42 | *Schindelman et al., 2011* |
| WWF Ecozones | | 43 | |

**Notes.**

1, http://aims.fao.org/agrovoc#.VG4QG_nF_ng; 2, https://code.google.com/p/behavior-ontology/; 3, https://github.com/tucotuco/bco; 4, https://code.google.com/p/biological-spatial-ontology/; 5, https://www.ebi.ac.uk/chebi/; 6, https://marinemetadata.org/references/cmecshabitat; 7, http://pantheon.generationcp.org/index.php?option=com_content&task=section&id=7&Itemid=35; 8, https://www.eagle-i-net/; 9, http://ecolexicon.ugr.es/en/index.htm; 10, http://explorer.natureserve.org/classeco.htm; 11, http://www.environmentontology.org; 12, https://marinemetadata.org/references/eunishabitat; 13, http://www.ebi.ac.uk/efo/; 14, http://www.obofoundry.org/cgi-bin/detail.cgi?id=exo; 15, http://www.pombase.org/; 16, http://wiki.pro-ibiosphere.eu/wiki/Traits_Task_Group; 17, http://www.yeastgenome.org/fungi/fungal_anatomy_ontology/; 18, http://bioportal.bioontology.org/ontologies/GAZ; 19, http://geneontology.org/; 20, http://www.geonames.org/; 21, http://www.getty.edu/research/tools/vocabularies/tgn/index.html; 22, http://www.gadm.org/; 23, http://www.human-phenotype-ontology.org/; 24, https://code.google.com/p/information-artifact-ontology/; 25, http://www.iucnredlist.org/technical-documents/classification-schemes/habitats-classification-scheme-ver3; 26, http://www.informatics.jax.org/searches/MP_form.shtml; 27, http://www.emodnet-seabedhabitats.eu/; 28, https://marinemetadata.org/references/cfregions; 29, https://semtools.ecoinformatics.org/oboe; 30, http://wiki.geneontology.org/index.php/Extensions/x-attribute; 31, http://obi-ontology.org/page/Main_Page; 32, http://microbialphenotypes.org/wiki/index.php/Main_Page; 33, http://obofoundry.org/wiki/index.php/PATO:Main_Page; 34, http://planteome.org/amigo/cgi-bin/crop_amigo/browse.cgi?; 35, http://www.plantontology.org/; 36, http://planteome.org/amigo/cgi-bin/crop_amigo/browse.cgi?; 37, https://github.com/PopulationAndCommunityOntology/pco; 38, https://github.com/oborel/obo-relations; 39, https://sweet.jpl.nasa.gov/; 40, http://www.sequenceontology.org/; 41, http://uberon.github.io/; 42 http://www.wormbase.org/; 43, http://wwf.panda.org/about_our_earth/ecoregions/ecoregion_list/.

## Using phenotype and environment ontologies in ecology
### Coping with climate change in conservation, management, and agriculture

*Example question.* Which species or crop varieties are projected to do well in my locality over the next century?

*Background.* Climate change is anticipated to affect environmental conditions with unprecedented speed. Knowledge concerning the responses of ecological communities to these changes is very limited: adaptation and migration are among numerous possibilities that must be considered. Conservation and agricultural resources are also limited, so identifying and focusing interventions on taxa that are less able to adapt can be very helpful. Besides commonly-used projections based on species distributions models, another strategy for identifying at-risk species is to assess their vulnerability based on their traits (i.e., phenotypes). By linking phenotypes to specific environmental conditions, taxa or strains that are likely to thrive (or not) under those conditions can be identified. For example, cataloguing phenotypes that are more prevalent among organisms that live in hot and wet environments and detecting their presence in organisms whose environments are warming and becoming more humid, allows some bearing on the later organisms' ability to cope with such climate change. In agriculture, this can be used to identify crop varieties that are likely to give higher or more stable yields under specific conditions or wild relatives of crops that may possess useful traits. In conservation, similar analyses can identify species at risk of extinction (*Thormann et al., 2014*). One system that hints at performing this type of analysis currently is SemanticWildNET (*Henderson, Khan & Hunter, 2007*), which links data about birds and snakes to environmental conditions in Australia.

*Current workflow.* Steve works for a seed company that serves the southern Great Plains in the USA. General Circulation Models (GCMs) project that over the next 25–30 years

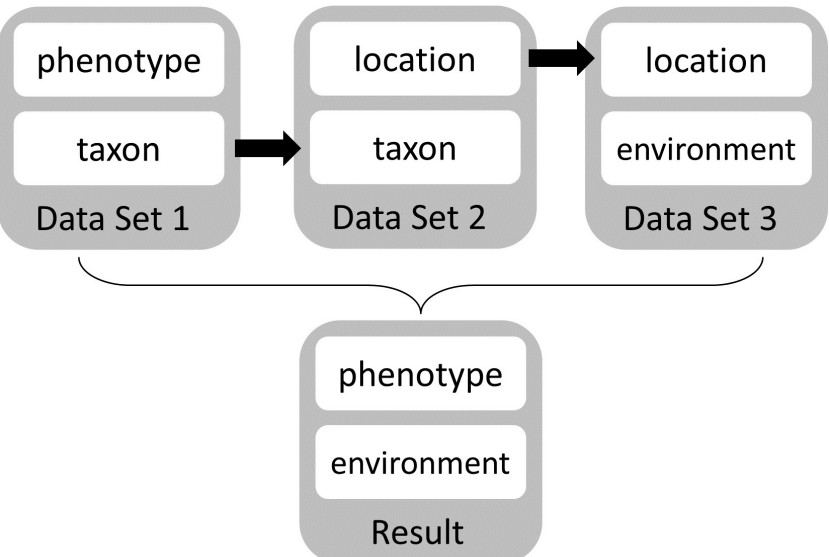

**Figure 2 Manual workflow conceptual diagram.** This diagram shows the manual workflow to link phenotype and environment data sets using current tools and services.

farmers in the southern Great Plains will experience drier, warmer, and longer summers. Steve's company wants to develop sorghum hybrids that perform well in drier, warmer climates—both to enable expansion into current marginal locations and to accommodate future environmental conditions. His company has spent three years measuring sheath reflectance via unmanned aerial vehicles as an indicator of cuticular wax constitution, a trait that is associated with moisture retention (*Jenks et al., 1992*; *Jenks et al., 2000*). Steve and his colleagues subsequently selected and carefully analyzed wax constitution as well as other drought tolerance indicators (e.g., leaf curling, days to maturity, etc.) for genetically diverse potential parental lines that vary in drought tolerance in different environs. Steve finds additional data sets that link geolocations and associated environmental conditions to taxon phenotypes for crops (e.g., GRIN, the Germplasm Resources Information Network; http://www.ars-grin.gov/npgs/) and link taxa to phenotypes (e.g., the TRY Plant Trait Database, https://www.try-db.org/; *Kattge et al., 2011*), but these resources have only minimal useful data. He also accesses environmental data sets that include information about weather (NOAA), soil (*USGS*) and climate projections on a 1 km spatial grid (WorldClim data set, *Hijmans et al., 2005*) through publicly available, online sources. Steve decides that the best strategy for attaining outstanding hybrid performance for warmer, drier environments is to manually link phenotypes of promising novel germplasm to environmental conditions using the taxon name and location (geographic coordinates) as a metadata bridge (Fig. 2).

Much of Steve's data comes from proprietary lines. To work with his data alongside relevant public information, he must download all files of interest to a local machine. Because he lacks programming skills, he must manually locate the specific data of interest from each data source and then make decisions about appropriate integration using

written documentation from the data provider. The data preparation and integration takes six months of full-time work. When Steve finally has his data ready to analyze, he must pick a statistical workflow and software package that can identify phenotypes and environmental conditions that are observed together. The next step would be to look at the climate projections to find the projected environmental conditions his customers are likely to be facing and use these to identify the ideal suite of phenotypes for that climate regime. The final step would be to identify germplasm accessions with desired traits.

*Future workflow: agriculture.* Steve works for a seed company that serves the USA. GCMs project that over the next 25–30 years farmers in the southern Great Plains will experience drier, warmer, and longer summers. His company wants to start developing sorghum hybrids that will perform well in such conditions. Steve's company has bred and phenotyped a genetically diverse range of potential parental lines that differ in drought tolerance-associated phenotypes, and these lines have been annotated using ontology terms for traits (e.g., TO and PATO, Table 1) as well as corresponding growth conditions (EO, Table 1). Additionally, the habitat/environment of each line (or its source organism) is described by classes from an environment ontology (ENVO, Table 1). Steve queries his company's internal, semantically aware database for annotated records corresponding to the lines his company has developed that perform well in warmer climatic conditions and when subject to drought. This gives him a list of candidate lines (i.e., phenotypes and genotypes) to use in development of new hybrids. If the candidates seem unlikely to provide sufficient resistance to high temperatures and drought, he may choose to query a database containing information on land races and wild relatives of sorghum where these germplasm accessions have been marked up with phenotype ontologies (*Oellrich et al., 2015*), along with average rainfall and temperature data from the natural habitat of each species and/or annotations describing their habitat using classes from an environment ontology. If necessary he could introgress genetic material encoding drought or high temperature tolerance from land races or wild relatives of sorghum into his breeding lines.

*Current workflow: wildlife conservation.* Lupita is a park ranger who manages a coastal wildlife sanctuary. Some of the species in her sanctuary are listed as threatened by the IUCN. According to the latest climate change projections, her sanctuary is going to be hotter and wetter in 50 years. She has limited resources to maintain the biodiversity in her sanctuary for the long term. She needs to identify which species might be at risk under the projected future climate regime. Lupita decides to spend the next year researching the literature on all the species in her park. After this work, she is able to identify five taxa that are likely to need extra conservation resources.

*Future workflow: wildlife conservation.* Lupita manages a coastal wildlife sanctuary that contains threatened species. According to the latest climate change projections, her sanctuary is going to be hotter and wetter in 50 years. She has limited resources to maintain the biodiversity in her sanctuary for the long term. After some thought, she decides to identify at-risk species by comparing the traits of the organisms in her park

with traits of organisms that do not do well in hot, wet, coastal environments. Lupita searches a semantically aware, publicly-available biology database and finds a list of traits for vertebrates whose habitats do not include hot, wet, coastal environments and a list of traits for vertebrates with habitats that do. Comparing differences between the two data sets gives a list of candidate traits which suggest a taxon would be vulnerable to the projected climate regime. Searching for these traits across the species in her sanctuary, Lupita identifies two species that are highly likely to fare poorly in the projected climate, and she devotes resources to their conservation.

*How ontologies can help.* These types of queries, relating biological features to environmental conditions could be performed now using lengthy relational database queries, if the relevant databases existed. Ontologies can streamline the query by creating OWL definitions such as the one below for the wildlife sanctuary use case:

HotWetCoastal EquivalentTo (partOf some coast) AND (hasAvgAnnPrecip some (hasUnit value cm and hasMagnitude min 125.0)) AND (hasTemp some (hasUnit value C and hasMagnitude min 20.0))

Then an endpoint could be queried in this way:

```
SELECT ?phenotype WHERE {
  ?taxon hasPhenotype ?phenotype ;
    hasHabitat [a  HotWetCoastal]
}
```

This would give a collection of phenotypes present in hot, wet, coastal environments that Lupita could cross-check against the taxa in her park. However, because of the nature of the natural world, a list resulting from the strict Boolean logic of SPARQL/SQL/OWL is likely to be less useful than a ranked list of phenotypes with $p$-values, similar to a gene enrichment analysis for phenotypes (e.g., *Submarmanian et al., 2005*). Many tools exist for performing enrichment analyses, including R packages that use the Gene Ontology, such as Bioconductor (*Gentleman et al., 2004*). These types of analyses require large, annotated data sets that can only be built with the cooperation of many researchers.

*Challenges today.* A large proportion of phenotype and environment data are part of the "long tail of dark data" (*Heidorn, 2008*) that are not currently digital or discoverable. Although some phenotype, environment, genotype, and climate projection datasets are available, they are difficult to find and interconnect. These types of datasets can be cross-linked using space, time, or taxon, but the formats of the different datasets can pose a challenge to integration (e.g., *Reed, White & Brown, 2003*). In addition, metadata across disciplines, data types, and time periods are rarely consistent. Key data items used for integration, such as taxonomic names, change over time and lead to poorly linked data (*Edwards et al., 2011*; *Giles, 2011*; *Page, 2008*; *Franz et al., 2015*).

## Using phenotype and environment ontologies in taxonomy
### Connecting specimen phenotypes to environment

*Example question.* Which traits are common to or vary the most in beetles collected from deserts?

*Background.* Natural history collections worldwide contain approximately two billion specimens across all taxonomic groups (*Ariño, 2010*). Tens of millions of these specimens have their phenomes at least partly described in the form of published taxonomic descriptions (*Deans, Yoder & Balhoff, 2012*) and the vast majority have locality data and possibly even environmental data recorded on the specimen label or in a field notebook. Much of these data are not readily available for research, i.e., they have not been digitized. Connecting specimen-based phenotype data to environmental information that describes where the specimen was isolated can support predictive modeling of diversity and distribution.

The ongoing, massive digitization effort applied to collections is primarily done manually, though efforts are being made to automate where possible (*Barber, Lafferty & Landrum, 2013*; *Ellwood et al., 2015*). The environmental data associated with a specimen, typically a note on a specimen label, is typically transcribed verbatim. If a curator wants to annotate a specimen with an environment or habitat type or other metadata, the process of reading the information and relating it to an ontology class is entirely manual. This time-consuming workflow involves reconciling synonyms and disambiguating homonyms. Many of these manual tasks could be made more efficient with the assistance of computers and curators intervening only periodically.

A semantic model for representing specimen phenotypes has been developed (*Balhoff, Yoder & Deans, 2011*) and applied to taxonomic descriptions (*Mullins et al., 2012*; *Balhoff et al., 2013*; *Mikó et al., 2014*). This model applies phenotype statements directly to specimens. Each specimen, residing in an institutional collection, is associated with collecting event data, including where, when, how, and by whom it was collected. The "where" data could be connected to environment types and other environmental data through semantic annotation using environment ontologies.

*Current workflow.* Kate developed a set of research questions about how phenotypes exhibited by beetles—cuticular texture, hairiness, and color, for example—vary in desert environments. To test these hypotheses on a large scale, she would first need to find, read, and interpret descriptions of the beetle species suspected to inhabit deserts. Kate does discover some recent beetle descriptions that are supplemented with semantic phenotype annotations (*Balhoff et al., 2013*; this approach is in the works for some beetle taxa (A Smith, pers. comm., 2015)), which allow for computation. Simple queries, like "show me all the texture (PATO:0000150) phenotypes," allow for relatively rapid, precise discovery of relevant phenotypes, but very few descriptions currently have data in this format. Using her brain, she begins to pull relevant phenotypes from older literature: "pronotum rugose," "elytra striate," "elytra with parallel ridges oriented anteroposteriorly," "punctate areas

present on femur and tibia." Given the lack of standard terminology and syntax, this first set of texture phenotypes takes a long time to assemble.

Kate's next step is to verify that these beetles are indeed desert-dwellers. She reads through the dense Material Examined section of a taxonomic treatment and tries plotting collecting events on a map. Using GIS desert-polygons she begins to narrow the taxa and specimens of interest. She then pulls specimen data from the Global Biodiversity Information Facility (GBIF), but there still aren't enough data to robustly test her hypotheses. She now must visit collections and digitize and interpret collecting event data from tiny paper labels affixed to specimens of her target taxa. Localities must be georeferenced and then interpreted as being desertic or not. If they *are* desertic it might be useful to apply fine-grained annotations—i.e., what kind of desert the habitat is. Kate recognizes that an ontological approach would facilitate analyses at different scales and so she employs ENVO classes: cold desert, rocky desert, stony desert, sandy desert (each is a subclass of desert). For beetles collected in the foothills of the Sandia Mountains (New Mexico, USA), she requests a new class: high desert. The lack of descriptive standards, digitized specimens, and user-friendly tools, make the research impossible to complete in any large-scale way.

*Future workflow.* Kate develops the same set of research questions described above: how do certain phenotypes vary in beetle species that live in deserts. Taxonomists are now generating all descriptions and redescriptions using a semantic approach. Anatomy terms are pulled from ontologies and combined with phenotype descriptors from PATO, views from the Biological Spatial Ontology (BSPO, *Dahdul et al., 2014*) and other relevant ontologies to make computable phenotypes that are applied directly to specimens. She can recover relevant phenotypes (and the associated specimens) using simple queries.

Tools have also been developed that facilitate habitat characterization at the time of collection and/or specimen digitization. All newly accessioned specimens in a collection are assigned to environment classes in ENVO, which is increasingly refined and expanded. For older specimens, which are still being digitized, the collection database application can offer suggested classes or even rule out certain environments, based on millions of accumulating data points. Specimens with relatively vague locality labels, like "State College, Penn." or "NY" (Pennsylvania and New York, USA, respectively), for example, could have come from a number of habitats: deciduous forest, pine needle litter, riparian, turfgrass, lab cultures, etc. Based on prior records from those localities, they are very unlikely to be from deserts. Therefore, Kate opts to ignore those data for her analyses.

*How ontologies can help.* In this use case, Kate's goal is to assemble data that are relevant to her research question on phenotype correlations with environment. One way ontologies can help is by providing a structured and controlled vocabulary of well-defined, machine-readable terms that taxonomists can use in annotations on collecting event data. The formal logic inherent in ontologies also allows Kate to find relevant data. A single query—Show me all the texture phenotypes for beetles—would yield phenotypes annotated with types of textures, like wrinkles, rugose, punctate, striate, and smooth and
the specimens with which they are associated. One more query—"of the beetle specimens with those phenotype annotations, show me all the specimens collected in deserts" (which includes high deserts, sandy deserts, stony deserts, etc.)—yields all the data available for her subsequent evolutionary analysis.

*Challenges today.* Inconsistencies in geographic metadata associated with specimens are a major roadblock in connecting phenotypes and environments (*Vollmar, Macklin & Ford, 2010*). Specimen metadata are filled with ambiguous and synonymous terms with inconsistent granularity. For example, the Plant Bug Inventory project database (http://research.amnh.org/pbi/; *Schuh, 2012*) uses thousands of habitat names to describe the localities where insect species were collected, including "cloud forest," "cloud forest with bamboo" and "cloud forest: oak trees, fern" (G Zhang, pers. comm., 2014). The documentation required to relate these terms to each other is currently absent. In addition, high-level (but imprecise) locality information (e.g., "State College, Penn.") is quite common for museum specimens and cannot be associated with fine-grained environment types. Further, specimen labels often contain somewhat vague terms such as "neotropical" or "mesohaline" that correspond to broad ecoregional definitions. According to Wikipedia, mesohaline is defined as water that is between 5 and 18 salinity (https://en.wikipedia.org/w/index.php?title=Salinity&oldid=685611535; accessed 16 October 2015), but it is seldom clear whether a collector has intended a precise definition such as this when writing the label. Thus, associating many specimens currently in collections with well-defined environments may not be possible.

Specimen metadata often include a latitude and a longitude or a locality name, which may be used to infer the environment, but environments change over time. For example, a specimen may have been collected from a desert, which has since been paved over in the expansion of a metropolitan area. Environments are also subject to cycles such as seasonal, diel, or tidal. All of these factors make date and time important metadata. Annotating specimens in more three-dimensional environments, such as the ocean or a mountain plateau, requires yet another piece of information—depth or altitude.

## Using phenotype and environment ontologies in phylogeny
### Reconstructing ancestral features and habitats

*Example question.* Do species that have independently reduced or lost their eyes share common environments now or in the past?

*Background.* To infer the most probable features of a common ancestor given a phylogenetic tree and the phenotypes of extant species, researchers utilize several well-developed parsimony and likelihood methods. Similarly, the habitat preference of living species can be used to reconstruct evolution of ecological niches. Connecting the phenotypic data from species with their habitat and environmental data allows efficient analysis of these associations, allowing, for example, the disentangling of evolutionary adaptation from other causes of phenotypic convergence.
Current methods of ancestral reconstruction rely on the uniform identification of a limited number of environmental traits (e.g., habitats). Users have parsimony, likelihood, and Bayesian methods at their disposal for ancestral state reconstruction (e.g., Mesquite, *Maddison & Maddison, 2014*; BEAST, BayesTraits, and R packages such as ape). These methods allow for both discrete and continuous values. For discrete characters, ancestral states are calculated from the specific character states (e.g., environmental traits) found in the species. For example, for a clade of species that live in either "deep sea" or "underwater cavern" habitats, ancestral state reconstruction is limited to these discrete habitats, i.e., the ancestor can be hypothesized to have lived in either one or the other habitat. However, an ontology can show that "deep sea" and "underwater cavern" are both subtypes of an "aphotic marine environment", and thus this parent term reveals this as a potential ancestral state for this clade.

*Current workflow.* Jane examines museum specimens of organisms belonging to a clade of freshwater fishes which encompasses several hundred species. She discovers that the eyes vary in their level of development: completely absent in some species, reduced in others, and fully developed in most. After mapping this trait on a well-supported phylogeny, she concludes that eye reduction and loss has occurred independently several times in this clade. This leads her to hypothesize that the changes in eye development are associated with a species' habitat. She goes to the museum databases and finds that the original descriptions of the collection sites for these specimens are recorded as free text in the Darwin Core field "verbatimLocality" (http://rs.tdwg.org/dwc/terms/verbatimLocality). She enters the "verbatimLocality" data into her matrix of features mapped onto the phylogeny. Jane notices that several terms might be synonymous and begins to research the specific definitions of the terms used and does her own research into conditions at each locality. After one month of reconciling locality terms, she begins to notice that species with reduced or absent eyes are all from subterranean environments. She proceeds with her study, now examining other environmental factors or phenotypic traits that might play a role in their shared habitat type.

*Future workflow.* While examining several hundred museum specimens of organisms belonging to a clade of freshwater fishes, Jane discovers that the eyes vary in their level of development. Mapping this trait on a well-supported phylogeny shows that eye reduction and loss has occurred independently several times in this clade. This leads her to hypothesize that the changes in eye development are associated with a species' habitat. She goes to the museum databases and finds that the original description of the place from where these specimens were collected was recorded as free text in the Darwin Core field "verbatimLocality" (http://rs.tdwg.org/dwc/terms/verbatimLocality), but the text is mapped to classes in an environmental ontology such as ENVO. She downloads these classes for all species and adds them to the matrix of features that are mapped to the phylogeny. She sees that species with reduced or absent eyes are from localities variously described as "shallow pool in cave," "deep water well," "deep phreatic habitat," and "swallow hole." A visualization tool allows her to see the ontological classes which

these descriptions have been mapped to as well as any shared hierarchies or relations to other classes. She notices that these descriptions share "groundwater" as environmental material and "subterranean" as an environmental quality. She proceeds with her study, now examining other environmental factors or phenotypic traits that might play a role in their shared habitat type.

*How ontologies can help.* Text descriptions of environments and habitats are very heterogeneous in the terms that are used and the granularity applied. In the current workflow of this use case, Jane spends a month examining habitat descriptions and grouping them in a way that is useful for her analysis. An ontology that describes the highly descriptive environments as subtypes of broader classes, in combination with a visualization tool for ease of use, would allow Jane to quickly find the broader classes "subterranean" and "groundwater". In addition, the same ontology could allow a query for a broader class, such as "subterranean" that would find habitats that are subtypes of "subterranean" or part of "subterranean."

*Challenges today.* As in the other use cases, environmental ontologies must be provisioned to include the classes relevant to a broad range of habitat types. Additionally, and similar to other use cases, phenotypes of taxa that are represented in a computational format must be readily available. The challenge unique to this use case is that methods of phylogenetic optimization that utilize ontological relationships need to be developed. This will require consideration of the hierarchy of class relationships such that the semantic similarity (*Pesquita et al., 2009*; *Resnik, 1999*) among differing ancestral states at a particular node is taken into account when calculating the appropriate assignment of a state to that node. Further, visualizations of the distribution of phenotypic and environmental features on the tree that display, e.g., the most similar ontological parent classes across nodes, need to be developed. A prototype to create an ancestral phenotype ontology has previously been made by *Ramírez & Michalik (2014)*.

## Using phenotype and environment ontologies in behavioral ecology
### Including species interactions in habitat assessments

*Example question.* How will this predatory wasp affect the spider population in my vegetable garden?

*Background.* Behavior is a phenotype that can be influenced by the presence or absence of other organisms. The presence of other taxa can be just as important as abiotic features for determining suitability of an environment for habitation by members of given species. An observation of a taxon exhibiting a stress behavior is very different from an observation of the same taxon exhibiting a feeding behavior. Changes in the ranges of organisms due to climate change or accidental introduction is another way by which environments change and become more or less suitable for specific phenotypes, such as feeding or courtship

behaviors. These behaviors are very important and when they are interrupted, can increase or decrease abundance of the affected organism.

Current methods for retrieving behaviors that might be predictive of species interactions mostly rely on published or unpublished ethograms and incidental comments in taxonomic descriptions or experimental studies. There are databases of species interactions (*Poelen, Simons & Mungall, 2014*), but these reflect interactions observed and reported in the literature, without the behavioral content to make predictions about possible interactions resulting from the introduction or range expansion of one or both species. Ideally, behavioral descriptions would include specific environmental preferences as well as details of foraging, anti-predator, and courtship behavior. The ability to make predictions of interactions would be an important contribution when considering planned introductions or when setting priorities for preventing unintentional spread.

*Current workflow.* Larry depends on his vegetable garden for food and on the spiders within it for pest control. He frequently sees the jumping spider, *Phidippus clarus Keyserling, 1884–1885*, in the garden. *Phidippus clarus* is a widespread and common spider in the Eastern US (*Edwards, 2004*) and has been demonstrated to be capable of controlling an experimental population of herbivorous insect pests (*Hoefler, Chen & Jakob, 2006*). Larry hears from his local agricultural extension office that a South American wasp that preys on spiders has been accidently introduced nearby. Should Larry be concerned that the presence of the wasp will lead to more pests in his garden? Larry takes the day off work to go to the local university library and asks a librarian to help him find information about *P. clarus* and the South American wasp. Much of the information he needs is in table format (ethogram) or in narrative text (comments in taxon descriptions and experimental studies) and is difficult to decipher. The librarian makes him aware of a database of species interactions that is easier to understand, but no data for *P. clarus* are available. At the end of the day, Larry is still uncertain about the effect of the wasps on his garden spider population.

*Future workflow.* Larry learns of an accidental release of predatory wasps on the jumping spider population in his vegetable garden. Should he be concerned that the presence of the wasp will reduce the spider population and lead to more pests in his garden? Larry checks a gardening app on his mobile device that uses a combination of ontologies and observation data to power a Q&A engine about nature in his area. Through a simple user interface, he asks the app if the wasp is likely to affect the jumping spider and whether there are additional potential consequences. Guided by the ontological structure available in its back-end, the app states that (1) *P. clarus* is known to spend large amounts of time on the tops and tips of plant shoots, and commonly lays its egg sacs near the tips of shoots (*Edwards, 1980*; *Hill, 2014*), and (2) the wasp searches for prey on the tops and tips of plant shoots. The inference engine used by the app is able to predict that the introduced wasp is likely to interact with Larry's population of *P. clarus* spiders. Because of where eggs are laid, this would have the potential to interrupt *P. clarus* reproduction and thus reduce pressure

on his garden pests. With this information, Larry spends an hour making several wasp traps out of old plastic bottles to place in his garden.

*How ontologies can help.* The key to recognizing the threat from the wasp is identifying the spatial intersection between where *P. clarus* spends its time (alternatively, its habitat preferences) and where the wasp searches for potential hosts. This relies on a having a common vocabulary (which would likely blend a micro-environment ontology with a plant gross anatomy ontology) that can express the preferences of both the spider and the wasp. Secondarily, a representation of the host preferences of the wasp (broad taxonomic range, possibly with specific exclusions, and phenotype features such as a size range) facilitates the step from spatial overlap to host selection. Finally, knowledge of the spider's defense behaviors (e.g., biting, kicking, and dropping to the ground) could either strengthen or weaken the predicted threat. Since Larry's interest is in biocontrol, not jumping spiders in particular, the app would likely need to apply this same reasoning to all known or predicted pest control predators in the garden.

*Challenges today.* Environment ontologies currently do not explicitly incorporate species interactions in the definition of their classes; however, an ontology for describing experimental conditions (EO) does describe interactions between plants and other organisms in their environment. Many environment ontologies, as they are currently structured, may not capture features relevant to whether an environment will support specific behaviors, which can be very important data. Not all taxa will engage in important behaviors in all environments, thus for many studies, presence/absence data are not adequate. Creating a new set of ethological ontologies and developing relations from their classes to those present in environmental ontologies has great potential to address these issues, but requires significant effort to realize and maintain.

## DISCUSSION

### Challenges

The process of developing the use cases identified several major barriers to linking phenotype and environment. These fall into two categories: challenges of coverage and challenges of interoperability.

### *Challenges of coverage*

*Variable granularity.* Environmental data are reported with varying degrees of granularity that can take the form of nested categories (e.g., continent—country—province—township—street), intervals (e.g., ±30 km), or significant digits (e.g., 5.236 vs 5.2). Some data sets, especially species observations, include highly granular metadata specifying the exact location or exact conditions under which a specimen was collected (such as collecting an insect from under tree bark or collecting an organism in the presence of its predator). Although existing ontologies cover many of the scales of interest, gaps prevent sufficient detail to capture all of the environmental data provided in connection with collected specimens or published studies. These are critical for some taxa, such as insects collected

from under bark (*Jain & Balakrishnan, 2011*). Currently, such data are not discoverable due to the paucity of terms in existing ontologies and the lack of easy-to-use tools that allow for semantic annotation with multiple ontologies.

*Terms and definitions.*  One of the biggest challenges in creating ontologies for application to disciplines that have a long history of published knowledge is the translation of the information in human-readable narrative into a machine-readable form. Human language is very difficult for a machine to understand largely because of its variability and nuance. Different terms (i.e., synonyms) can be used to refer to the same concept, while a single term (i.e., homonyms) can refer to multiple, different concepts. The human brain copes with this uncertainty by understanding context. One way for a machine to cope with the variability of natural language is to provide it with an ontology that includes synonymous terms; however, this can be difficult to maintain because language evolves rapidly. Homonyms are a word-sense-disambiguation problem, which requires heuristics about context to infer meaning; it is an active area of research (*Zhan & Chen, 2011*). A homonym example applicable to environments is the term "scrubland," which means something very different in California and South Africa. In this case, significant disambiguation could be achieved by cross-referencing terms with geo-location or place names using resources such as GAZ (*Buttigieg et al., 2013*), a gazetteer developed along ontological principles.

*Incomplete ontologies.*  The development of ontologies in the biodiversity sciences has grown rapidly but is relatively new, thus coverage is still small (Table 1). The OBO Foundry Library (http://www.obofoundry.org/), a repository for biological ontologies, contains 22 ontologies relevant to environments and phenotypes, with a total of 136,480 classes. Of these ontologies, only one describes environments (ENVO) and one describes plant environmental conditions in experimental treatments (EO). Eleven are phenotype or anatomy ontologies that cover specific taxonomic groups, such as fungi (FYPO), animals (UBERON primarily for Chordates, with other ontologies such as PORO for specific clades like Porifera (*Thacker et al., 2013*)), and plants (TO) (Table 1). Many other taxa, such as the microbial eukaryotes, do not have dedicated ontologies. Furthermore, existing ontologies lack many key concepts required for application to the many facets of biodiversity. This argues for the need of "living" ontologies (actively maintained and highly responsive to user requests) that can be updated continually and with tools and services to allow users to request new classes and update existing classes with low overhead. Ontology development is extremely time-consuming (*Dahdul et al., 2015*), and it must be driven by scientific requirements, not by attempts to fully provision them *a priori*. Further, provenance, i.e., the record of authorship involved in term development through persistent digital identifiers such as ORCID (orcid.org), is a poorly-developed feature in most ontologies, though important for providing credit to contributors.

### Challenges of interoperability

*Data integration.*  Linking environments, locations, and phenotypes will require interoperability between several data types with the varying granularity used in biodiversity and

geoscience. These include data types from political and physical geography, coordinate systems, gazetteers, as well as representations of environment and habitat. GeoNames has linked political geography and some physical geography with coordinates (http://www.geonames.org/). A specimen with coordinates can easily be linked to any number of political entities using the GeoNames API. The same has not been accomplished for habitats; however, the components required to accomplish this are falling into place. For example, the LifeMapper (*Prajapati, 2009*) and Map of Life (*Jetz, McPherson & Guralnick, 2012*) projects use ecological niche modeling to map species distributions based on environmental conditions. Additionally, the Encyclopedia of Life TraitBank (http://eol.org/info/516) links taxa to their habitat type and phenotypic traits, but not to geographic coordinates (*Parr et al., 2015*). Once greater ontological representation of the link between species and their environments is accomplished, robustly linking species' phenotypes to their environments and locations become readily achievable.

In addition to spatial variation, environments show considerable variation over time and often change over daily and seasonal cycles. This makes temporal data a key component for meaningful integration. Environmental conditions measured at 14:00 can be very different from those measured at 02:00 in the same location. The measurements made at the former may not apply to a specimen collected at the latter. In addition, an organism is rarely only exposed to conditions measured at a single place and time. Some degree of integration is required to get a complete picture of an environment associated with a phenotype (referred to as the "exposome" in epidemiology, *Wild, 2005*).

*Knowledge representation for data integration.* Regardless of limitations in domain ontology (e.g., environment, phenotype) coverage, methods for linking phenotypes to environments using existing ontologies do exist (Figs. 3 and 4). The Extensible Observation Ontology (OBOE) provides an extensible knowledge graph for linked measurements and has been described in detail elsewhere (*Madin et al., 2007*; *Madin et al., 2008*). Briefly, the fundamental OBOE model is built around an "observation" class which is an observation of an "entity" and has one or more "measurements." Observations can also have a context of other observations. Phenotypes and environments can be linked by representing an organism observation with a location observation as its context (Fig. 3A). OBOE can model categorical and numerical measurements (Fig. 3B). Thus, a geolocation, a data point, or a country code can be added to a location observation that provides context for an organism observation. OBOE allows the use of literals as instances, meaning a measurement can have as a value a string or a URI, which can be helpful when a needed URI does not exist.

Although OBOE is well suited for describing observations, it was not originally built to manage information about specimens or taxa. The Biological Collections Ontology (BCO) (*Walls et al., 2014a*; *Walls et al., 2014b*; *Deck et al., 2015*) offers an alternative way to link data, based on ontology design principles from the Ontology for Biomedical Investigations (OBI; *Brinkman et al., 2010*), adapted for biodiversity science. A key element of BCO is the difference between a specimen collection process, which has a material entity

(i.e., specimen) as output and an observing process, which has data as output. *Deck et al. (2015)* describe how information about locations (e.g., coordinates or environmental conditions) and taxonomy (e.g., the identification process or species name) can be linked to specimens. A similar approach can be used to link phenotypic data to observations of organisms in their environment. At its most basic, the BCO (via OBI) represents the observing process as a type of assay (an OBI class). Rather than representing taxonomic information as an observation, BCO has a class for taxonomic identification process, which, like assay, is a subclass of OBI:planned process (Fig. 4A). Figure 4B shows how the same data from Fig. 3B would be mapped to instances of BCO classes. OBOE and BCO were developed for different use cases (managing ecological observation data for OBOE and integrating biodiversity data that follow different metadata standards for BCO) and therefore have different approaches to representing observations. Nonetheless, there is significant overlap between the two ontologies (e.g., OBOE's observation corresponds closely to BCO's observing process), and ongoing efforts to align them are likely to lead to a harmonized model that can work for many different use cases, including those described above and for more complex data systems that could support work similar to the proof-of-concept demonstrations described below.

*Ontology legacy alignment.* The development of successful ontologies is often driven by a "bottom-up" community approach. While this results in a product that is relevant for users, it can also result in multiple partially overlapping ontologies, despite efforts to prevent duplication (e.g., *Smith et al., 2007*). For improved integration and inferencing, overlapping ontologies need to be properly aligned and those alignments need to be maintained over time. If not done properly, inferencing may be inhibited or precluded altogether. This is a general problem that is not unique to environment or phenotype ontologies (*Cregan et al., 2005*). A "top-down" approach to ontology development, in which classes that constitute the top levels of a new ontology come from an existing domain or upper-level ontology (e.g., CARO, UBERON, PO, BFO—*Grenon & Smith, 2004*; *Haendel et al., 2008*; *Mungall et al., 2012*; *Cooper et al., 2013*), can result in a shared structure and homogenized development across ontologies, although more specific classes will still require alignment. Aligning ontologies manually is a large task and it is difficult to know the full consequences of an alignment without testing (*Ochs et al., 2015*). The ability to support the provenance of alignments and re-alignments can translate into trust and continued investment. Numerous semi-automated tools for alignment have been developed (e.g., *Granitzer et al., 2010*; *Chen et al., 2014*). Challenges include setting up proper relations between classes in different ontologies such that the logical outcomes are valid and consistent (*Franz & Peet, 2009*; *Meilicke & Stuckenschmidt, 2009*; *Jiménez-Ruiz et al., 2009*; *Jansen & Franz, 2015*; *Franz et al., 2015*). The time and effort spent on maintaining alignments and interoperability can be eliminated if shared community resources are instead developed (*Dahdul et al., 2015*). For example, several independent anatomy ontologies for vertebrates (teleost (*Dahdul et al., 2010*); amphibian (*Maglia, Leopold & Pugener, 2007*); vertebrate skeletal (*Dahdul et al., 2012*), and vertebrate

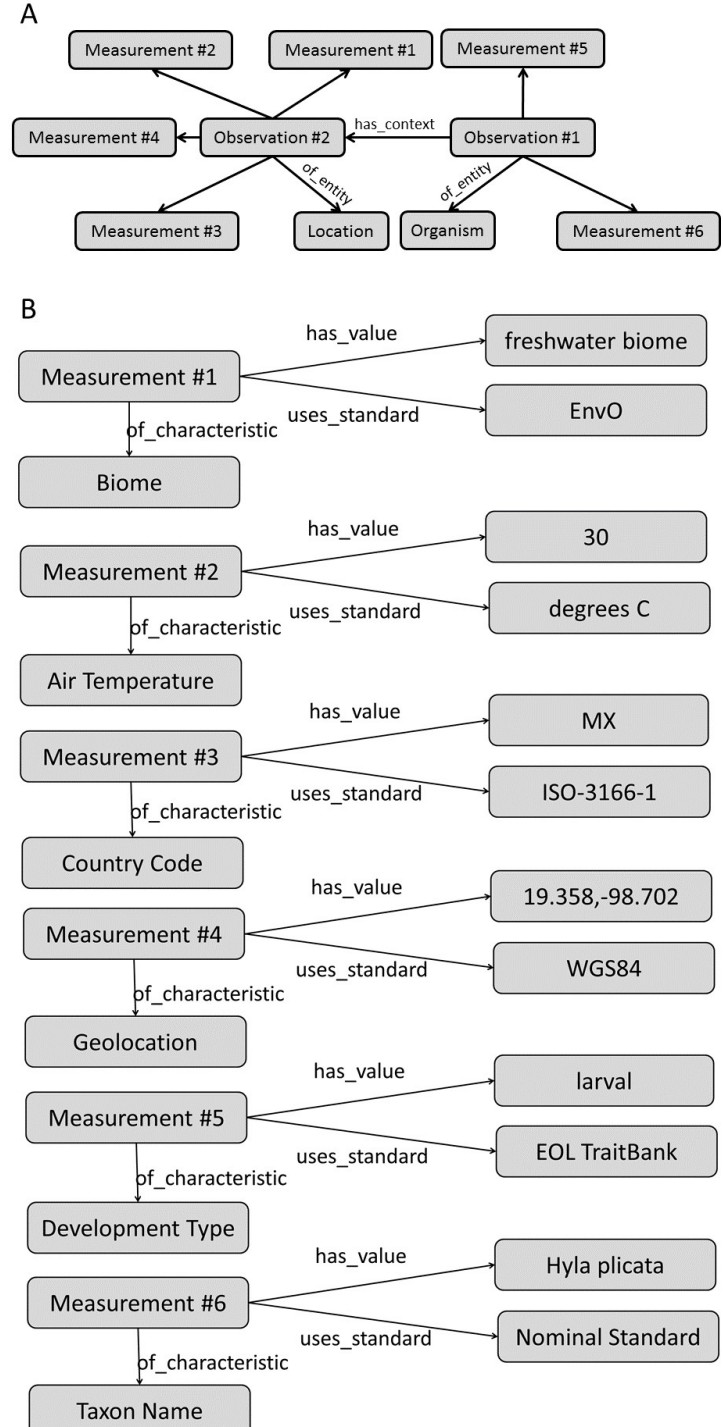

**Figure 3 Using the OBOE ontology to link phenotype and environment.** This demonstrates linking phenotype and environment using instances of the OBOE classes Entity, Observation, and Measurement. (A) Links between Entity, Observation, and Measurement OBOE classes. (B) Example measurements of phenotypes and environments using instances of the OBOE classes. Numbered measurement instances are consistent across A and B. This representation is simplified with regards to the taxonomic entities in play (*Baskauf & Webb, 2015*).

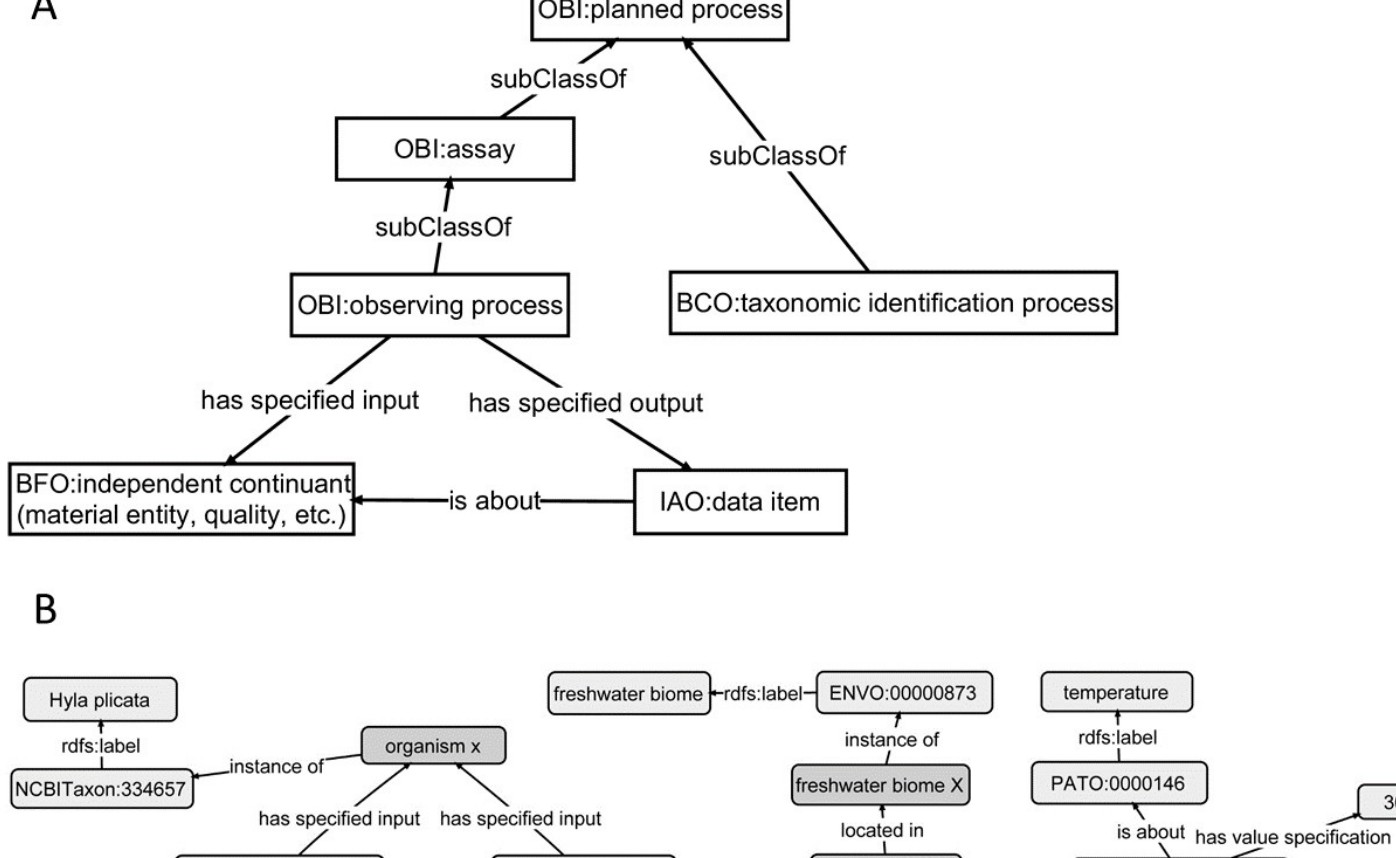

**Figure 4 Using the Biological Collections Ontology (BCO) to link phenotype and environment.** This demonstrates linking phenotype and environment using classes and relations from the Biological Collections Ontology (BCO). (A) A simple version of the classes and relations used to describe observations in the BCO, with classes imported from OBI (Ontology for Biomedical Investigations), IAO (Information Artifact Ontology), and BFO (Basic Formal Ontology). (B) Links among organism, phenotype, and environment, using the BCO model, using the same data as in Fig. 5. Light grey boxes represent either literal values (e.g., *Hyla plicata*), or instances of classes from external ontologies (ENVO, Environment Ontology; UBERON, Uber Anatomy Ontology; PATO, Phenotype Quality Ontology). Properties with a dwc prefix are imported directly from Darwin Core.

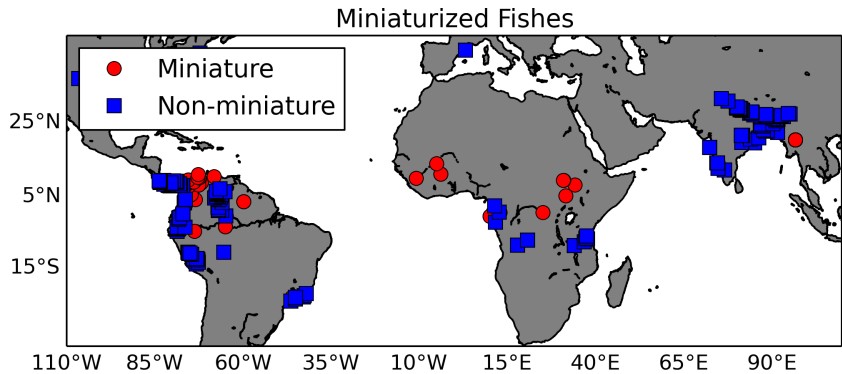

**Figure 5 Map of miniaturized fishes and their non-miniaturized sister taxa.** This map shows locations of fish species exhibiting the miniaturized phenotype (red circles) and their non-miniature sister taxa (blue squares). The georeferenced occurrence data were gathered from GBIF.

homologous organs (*Niknejad et al., 2012*)) were recently subsumed into UBERON, the metazoan anatomy ontology (*Haendel et al., 2014*), and new content and development is now focused on this single resource.

## Proof-of-concept demonstrations: linking environments and phenotypes
### *Miniaturization in fish*

*Question.* Has the evolution of miniaturization in fishes been driven by environmental variables?

Miniaturization is essentially the evolution of small body size and the associated set of phenotypes, typically reduction or loss of structures (*Hanken & Wake, 1993*; *Weitzman & Vari, 1988*). Authors have related this extreme change in body size to organisms whose habitats include highly acidic waters, typical of peat bog or black water habitats (*Kottelat et al., 2006*). As a proof-of-concept, we tested the hypothesis that miniaturization is correlated with environmental variables. Using a list of miniaturized fishes and their sister taxa extracted from the literature as input to GBIF (gbif.org), we created a list of 378 georeferenced observations from museum specimen records (http://doi.org/10.15468/dl.wzpgpe; Fig. 5). These species' latitude and longitude occurrence records were matched to the 1 km HydroSHEDS hydrography (*Lehner, Verdin & Jarvis, 2008*) using a horizontal distance tolerance of 3 km; they were then intersected with freshwater specific layers by *Domisch, Amatulli & Jetz (2015)*. In this data set, the watershed of each 1 km stream reach along the HydroSHEDS hydrography was delineated and then overlaid with climate (*Hijmans et al., 2005*), topography (*Lehner, Verdin & Jarvis, 2008*), land cover (*Tuanmu & Jetz, 2014*), and surface geology (*USGS*) layers. For this demonstration, the differences in two habitat variables, temperature and precipitation, between miniatures and non-miniatures were explored using a logistic regression (Table 2). A Wald test showed that the logistic regression had a $p$-value of 0.0025 with two degrees of freedom and a "goodness of fit" test gave a $p$-value of $9.4 \times 10^{-5}$ with two degrees of freedom. The

**Table 2 Results of a logistic regression comparing temperature and precipitation in the habitats of fishes with and without the miniaturization phenotype.**

| Variable | Coefficient | Std. error | $Z$ value | $p$-value | Confidence interval 2.5% | Confidence interval 97.5% |
|---|---|---|---|---|---|---|
| Annual mean temperature (°C) | $1.6 \times 10^{-1}$ | $5.3 \times 10^{-2}$ | 3.033 | 0.00242 | $6.6 \times 10^{-2}$ | $2.8 \times 10^{-1}$ |
| Annual mean precipitation (mm) | $3.4 \times 10^{-9}$ | $1.6 \times 10^{-9}$ | 2.139 | 0.03243 | $7.6 \times 10^{-10}$ | $7.1 \times 10^{-9}$ |

results showed that miniaturized fishes are found in warmer, wetter environments than their non-miniaturized counterparts. New data layers are being developed to test specific phenotypic hypotheses related to the habitats (e.g., pH, water flow) of miniaturized fishes (*Domisch, Amatulli & Jetz, 2015*). By annotating these habitat layers with environmental ontology terms (e.g., 'acidic water' annotated to the appropriate pH range), this data can be plugged into an ontology-based knowledge graph, such as the Phenoscape Knowledgebase (KB) (kb.phenoscape.org) which uses common phenotype ontologies to link evolutionary phenotypes to candidate genes from model organisms. This would enable large scale data aggregation and inferencing across habitat, taxa, and phenotypes, regardless of the natural language terms used, similar to use case number 3 above. The phenotypes of these taxa can then be retrieved in a phenotype × taxon synthetic supermatrix from the (KB) using the Ontotrace tool (*Dececchi et al., 2015*). The correlations of environment to ontology-based miniaturized fish phenotypes (e.g., mandibular sensory canal, absent; basibranchial 2 tooth, absent) can then be examined.

The necessary ontological structures to fully automate the queries above do not yet exist, but ontologies can improve this workflow in several ways. First, an ontological class representing the miniaturized phenotype (miniPheno) can be created using EquivalentTo and the structures described in *Dececchi et al. (2015)*. The taxon occurrence data can be represented similar to the structures proposed in *Henderson, Khan & Hunter (2007)* and the HydroSHEDS data can be represented as grids linked to data using GeoSPARQL:
grid (hasExactGeometry some Geom (asWKT some WKT (Polygon(lat long, lat long, lat long, lat long))) AND hasAnnualPrecipitation some_value)
Then, triples can be created to match taxa to HydroSHEDS grids:

```
CONSTRUCT {?taxon hasGrid ?grid}
    WHERE {?organism hasIdentification ?taxon
        ?taxon hasOccurrence ?occurrence.
        ?occurrence atEvent ?event.
        ?event locatedAt ?location.
        FILTER (?location geof:sfWithin ?grid)
    }
```

Finally, a SPARQL query can be used to retrieve the precipitation data for environments where taxa exhibiting the miniaturization phenotype have been observed.

**Table 3** Some of the URIs used to describe amphibian breeding and development in TraitBank.

| Term | URI |
| --- | --- |
| Breeding habitat | http://eol.org/schema/terms/BreedingHabitat |
| Development mode | http://eol.org/schema/terms/DevelopmentalMode |
| Terrestrial habitat | http://purl.obolibrary.org/obo/ENVO_00002009 |
| Intermittent pond | http://purl.obolibrary.org/obo/ENVO_00000504 |
| Permanent pond | http://eol.org/schema/terms/permanentFreshwater |
| Freshwater stream | http://eol.org/schema/terms/freshwaterStream |
| Direct development | http://eol.org/schema/terms/directDeveloper |
| Larval development | http://eol.org/schema/terms/larvalDevelopment |
| Paedomorphic | http://purl.obolibrary.org/obo/HOM_0000029 |

```
SELECT ?AnnualPrecipitation WHERE {
    ?taxon hasPhenotype miniPheno
    ?taxon hasGrid ?grid
    ?grid hasPrecip ?AnnualPrecipitation
}
```

### Amphibian reproduction

*Question.* Which amphibians in my neighborhood are most likely to have their breeding disrupted if a plan to drain a pond (the single source of year-round, standing freshwater) is implemented?

The Encyclopedia of Life links environments associated with a given species' habitat and phenotypes indirectly through taxon names. These data can be accessed and downloaded via TraitBank (*Quintero et al., 2014*; *Parr et al., 2015*). TraitBank uses Uniform Resource Identifiers (URI), many from existing ontologies, as a controlled vocabulary for describing characters and character states to facilitate large-scale data integration (Table 3). This data set was built by extracting data from the literature and applying the relevant, fit-for-purpose URI. As proof-of-concept, we queried TraitBank for breeding environment and developmental mode in 278 amphibian taxa. A Chi-Square Test was used to test for independence between habitat and reproductive mode. The data suggested an important reproductive difference between amphibians in aquatic and terrestrial habitats (Table 4). Ninety-nine percent of the amphibians with direct development breed in a terrestrial habitat. Ninety-eight percent of the amphibians with larval development (tadpoles) breed in an aquatic environment. This links the "larval development" phenotype to the "freshwater" environment and the "direct development" phenotype to the "terrestrial" environment.

Ontologies play a minor role in this proof-of-concept as a controlled vocabulary, which was important considering the variety of terms used to describe breeding habitat in the literature. In this case, term management was performed by humans who divided the described habitats into the categories relevant for this analysis. Using ENVO terms would be a much more flexible way to describe these data that would better preserve the

**Table 4** Breeding habitat and developmental mode for 282 species of amphibians.

| | Larval | Direct | $df$ | Test statistic | $X^2_{0.95}$ |
|---|---|---|---|---|---|
| Freshwater stream | 30 | 0 | 3 | 270.9 | 0.352 |
| Intermittent pond | 28 | 0 | | | |
| Permanent pond | 59 | 2 | | | |
| Terrestrial | 2 | 166 | | | |

description in the literature and be easily managed into higher level groupings through the type propagation rule. For example, taxa with breeding habitats reported as "cloud forest" or "forest" would both be subclasses of "terrestrial." Thus, if these structures were in place, breeding habitats could be described to the level of detail reported in the literature and still easily queried using high level descriptors as in the example SPARQL query below. The results of this query would be a list of Amphibian taxa that breed in terrestrial environments and their developmental mode.

```
SELECT ?taxon ?developmentalMode
    WHERE {?taxon hasClass :Amphibia .
        ?taxon hasBreedingHabitat :terrestrial .
        ?taxon hasDevelopmentalMode ?developmentalMode }
```

These examples provide demonstrations of the value of linking phenotype to environment, demonstrate how these links can be made with existing tools, and describe how further ontology development could help. More complicated research questions are likely to require more nuanced linking for several reasons. First, phenotypes frequently vary within a species; one cannot assume that every member of a species has the same phenotype. In these two examples, we chose traits that were consistent across all members of a species (miniaturization and developmental mode). In the miniaturization example, this allowed addition of the GBIF query results to the Phenoscape Knowledgebase results. Second, an organism's life style (ambush predator, nocturnal frugivore, etc.) within an environment is deeply rooted to its phenotypic composition. For example, a visual predator in an environment with low-light conditions may have a large eye phenotype while a scavenger in the same environment may have a small eye phenotype. Trying to connect an eye size phenotype to this environment would have to be clarified by including the ecological role of the taxon in a given ecosystem. Third, scale can be important. Taxa of very different sizes can experience the same environment in very different ways. For example, a soil protist will experience a forest environment differently than a vascular plant. Despite these challenges, the highly simplified fish and amphibian examples above still demonstrate the results of linking phenotypes and environments with existing data and tools.

## Summary

Providing data structures that improve integration of biological data is necessary for efficiently addressing complex research questions. The link between phenotype and

environment is fundamental to research in taxonomy, ecology, and phylogenetics; its relevance extends to the biomedical domain. One way to create this link is through the use of extensible ontologies designed to work across different data types, such as OBOE or BCO in combination with ENVO and other trait ontologies. Despite recent advances, significant challenges remain. We recommend the following three steps to increase interoperability between phenotype and environment data:

### Make it easy to contribute to existing ontologies

The existing suite of ontologies is not adequate for linking phenotypes and environments across the tree of life. To address this, new classes need to be added to extend and improve existing taxonomy, phenotype, and environmental ontologies. Some ontologies have well-developed pathways for submitting new classes and editing existing classes and resources to respond to requests quickly (e.g., Gene Ontology), but frequently the social processes of validating ontologies are not a part of the ontology platform.

### Georeference environments with temporal considerations

Many taxon observations are accompanied by geographical coordinates, collection date and time, but lack adequate environmental descriptions. While services exist that can translate coordinates into a municipality, retrieving environmental information using geographic coordinates is not yet possible across the globe. In addition, because environments are dynamic, temporal information should be used to filter results. A service is needed that can take spatiotemporal information and return data concerning environmental conditions and ontology classes corresponding to environment types. Map of Life (*Jetz, McPherson & Guralnick, 2012*) can provide some data corresponding to coordinates in some areas, but ontology classes are not yet available.

### Organize research communities that share common resources

Ontologies rely on community support, driven by scientific questions, to be relevant. Communities of experts can be organized around workshops co-occurring at conferences and funded through programs such as the National Science Foundation's Research Coordination Network. Significant progress on discipline-specific ontologies has been made through the use of targeted workshops (e.g., *Yoder et al., 2010*).

**Glossary**

**Collecting event** The process of specimen collection that occurs at a specific time and place.

**Cyberinfrastructure** The technological framework of interconnected databases and computers across institutions that enable and support advanced, large-scale scientific research.

**Darwin Core** A standard reference of terms related to biological diversity, in particular taxa and their occurrences. Darwin Core was created to facilitate sharing of biodiversity information.

**GCM (General Circulation Model)** From Wikipedia: a general circulation model (GCM), a type of climate model, is a mathematical model of the general circulation of a

planetary atmosphere or ocean and based on the Navier–Stokes equations on a rotating sphere with thermodynamic terms for various energy sources (radiation, latent heat).

**Genotype** The genetic makeup or set of genes of an organism.

**Georeferenced** Observations or specimen collection records that are associated with locality information (e.g., latitude and longitude).

**Human readable** Information that is presented in a format that can be understood by a human.

**Inferencing** Performed by software programs ("reasoners") that deduce logically consistent statements implied by the entities and relations asserted in an ontology or database.

**Knowledgebase** A database of interconnected information.

**Machine readable** Information stored in a data format that can be understood by a computer.

**Machine Learning (ML)** A type of artificial intelligence in which software programs have the ability to learn (make decisions or data predictions) without being explicitly programmed when given new data.

**Meta-analysis** A statistical analysis of data that is combined from independently conducted research studies.

**NLP (Natural Language Processing)** Methods used in computer programs to understand and extract data from natural (human) language.

**Ontology** A set of defined terms (classes, concepts) and the relations between them that represent the knowledge of a particular domain. Terms in an ontology are related in a directed, acyclical graph.

**OCR (Optical Character Recognition)** Automated conversion of images of text into machine-readable text.

**OWL (Web Ontology Language)** The name encompassing the set of web-based languages used for ontology building supported by the World Wide Web Consortium (W3C) international standards body and based on the rules of formal semantics.

**Phenome** The entirety of an organism's phenotypic traits.

**Phenotype** One or more observable characteristics of an organism.

**Provenance** History of data and its place of origin.

**RDF (Resource Description Framework)** A family of World Wide Web Consortium (W3C) specifications originally designed as a metadata model and generally used to model information in knowledge management applications.

**Semantic** Of or relating to meaning or context.

**Semantic annotation** The act of adding (i.e., 'tagging') information artifacts such as images, free-text anatomical descriptions, or specimen collection records, with classes from an ontology or similar resource which represents their meaning in a machine-readable fashion.

**Specimen** A whole organism or part of an organism preserved in a collection.

**Taxonomic description** Natural language description of a taxonomic group, typically includes phenotypic characters such as morphology and behavior.

**URI (Uniform Resource Identifier)** A string of characters used to identify a resource that enables interactions with representations of the resource over the internet.

**Vocabulary** Flat list of terms that can be used to classify data. These terms are not explicitly related to one another.

## ACKNOWLEDGEMENTS

The scientific meeting from which this manuscript arose was organized and supported by the Phenotype Ontology Research Coordination Network (NSF-DEB-0956049). Any opinions, findings, and conclusions or recommendations expressed in this material are those of the authors and do not necessarily reflect the views of the National Science Foundation. The authors would like to thank the participants of the 2014 Phenotype RCN Workshop for insightful discussions and extended dialogue leading to this paper.

### Funding

The scientific meeting from which this review arose was organized by the Phenotype Research Coordination Network, which is funded by the US National Science Foundation, grant number DEB-0956049. PLB's work on this project is supported through the Micro B3 project, funded by the European Union's Seventh Framework Programme (Joint Call OCEAN.2011-2: marine microbial diversity—new insights into marine ecosystems functioning and its biotechnological potential) under the grant agreement no 287589. SD received funding from the German Research Foundation DFG (grant DO 1880/1-1). PJ and LDC received funding from the US National Science Foundation (NSF IOS:0822201, IOS:1127112, IOS:1340112). CDS received funding from the US National Science Foundation grant number DEB 1208666. RLW was supported by the iPlant collaborative as part of the National Science Foundation Award Numbers DBI-0735191 and DBI-1265383. CJM and SEL were supported by R24OD011883 and by the Director, Office of Science, Office of Basic Energy Sciences, of the US Department of Energy under Contract No. DE-AC02-05CH11231. PMM and WMD were supported through by Phenoscape project (NSF grants DBI-1062404 and DBI-1062542). The funders had no role in study design, data collection and analysis, decision to publish, or preparation of the manuscript.

### Grant Disclosures

The following grant information was disclosed by the authors:
US National Science Foundation: DEB-0956049, IOS:0822201, IOS:1127112, IOS:1340112, DEB 1208666.
European Union's Seventh Framework Programme: 287589.
German Research Foundation DFG: DO 1880/1-1.
National Science Foundation Award: DBI-0735191, DBI-1265383.

Office of Science, Office of Basic Energy Sciences: R24OD011883.
US Department of Energy: DE-AC02-05CH11231.
Phenoscape project NSF grants: DBI-1062404, DBI-1062542.

## Competing Interests

Suzanna E Lewis and Paula M Mabee are Academic Editors for PeerJ. Anne E Thessen is an employee of The Data Detektiv, Waltham, Massachusetts. Eva Huala is an employee of Phoenix Bioinformatics.

## Author Contributions

- Anne E. Thessen analyzed the data, wrote the paper, prepared figures and/or tables, reviewed drafts of the paper.
- Daniel E. Bunker, Pier Luigi Buttigieg, Laurel D. Cooper, Nico M. Franz, Pankaj Jaiswal, Carolyn J. Lawrence-Dill, Martín J. Ramírez, Chelsea D. Specht, Lars Vogt, Rutger Aldo Vos, Jeffrey W. White and Guanyang Zhang reviewed drafts of the paper.
- Wasila M. Dahdul analyzed the data, contributed reagents/materials/analysis tools, wrote the paper, reviewed drafts of the paper.
- Sami Domisch analyzed the data, contributed reagents/materials/analysis tools, reviewed drafts of the paper.
- Peter E. Midford wrote the paper, reviewed drafts of the paper.
- Christopher J. Mungall contributed reagents/materials/analysis tools, reviewed drafts of the paper.
- Ramona L. Walls wrote the paper, prepared figures and/or tables, reviewed drafts of the paper.
- Andrew R. Deans, Eva Huala, Suzanna E. Lewis and Paula M. Mabee wrote the paper, reviewed drafts of the paper, hosted workshop wherein paper was initially conceived.

## Data Availability

The research in this article did not generate any raw data.

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
