# Peer review of "Emerging semantics to link phenotype and environment"

_PeerJ, doi:10.7717/peerj.1470_

## Round 0.1 · original submission · Major Revisions

Well, to say the reviewers were split on this paper would be an understatement. You received the full range of possible recommendations from accept to reject with a major revision in between. Nevertheless, all the reviewers and I enjoyed your paper and thought it a valuable contribution to the literature. The major 'concern' seems to be about fitting the 'scope' of the journal. But the good folks at PeerJ assure me that despite what their website says, they are interested in expanding into occasional review papers, if they are well done and informative. I think yours fits that bill. So please disregard the reviewers' comments about 'fit' to the journal. You are expanding the journal! Please do pay careful attention to the other comments as I think they are very instructive and helpful. In particular, there are a number of comments about this paper not looking much like a 'review' either - rather a case study. I called it an opinion piece with some novel research. As such, perhaps consider a title change (even just deleting ': A review' and then clearly in the intro, as you've done in the abstract, let the reader know what you are up to (as suggested by one reviewer).

Reviewer 1 ·

Basic reporting

Although the title of the paper suggests that it is a review, it does not have the form of a typical review paper - it does not primarily consist of an exhaustive review of the literature on a particular topic. It also does not have the typical form of a data paper - it does not primarily present and analyze the results of experimentation. Rather, it makes a case that ontologies are an effective mechanism for answering questions about interactions between phenotypes and environment.

To make this case, the authors:
- briefly describe ontologies and how they are developed,
- cite examples where ontologies have been developed to describe resources related to phenotypes and environments
- present a number of use cases that illustrate interesting questions involving phenotype and environment and describe challenges faced in linking phenotype and the environment,
- provide two proof-of-concept illustrations, and
- suggest two ontologies that might be used as models for linking

Although the authors provide data and analysis in their two proof-of-concept illustrations, it does not appear that this paper falls into the category of "Research Article" as required by the PeerJ Aim and Scope. The article also departs significantly from the Standard Sections template described in the Instructions for Authors.

Experimental design

Because this paper does not seem primarily to report the results of original research, it doesn't really present an "experimental design". Rather, its purpose seems to be to demonstrate the value of ontologies for addressing questions about the relationship between phenotype and the environment. Throughout the paper, the authors make statements that can be summarized as falling within the following four categories:
1. there are interesting questions that should be asked about interactions between phenotype and environment
2. lack of standardization of terms and clarity of definition of terms hampers addressing these questions
3. inability to link resources related to phenotype and environment makes it difficult to acquire the data necessary to answer these questions
4. terms whose definitions include semantic information make them a more powerful and effective tool for computation.

Descriptions of research challenges

The paper does a good job of discussing interesting areas for future research. Each of the use case sections includes background and "challenges" sections that describes challenges and problems that need to be overcome in order to make progress towards understanding relationships between phenotypes and environments. The Discussion section additional areas that impede progress: variable granularity, terms and definitions (i.e. disambiguation problems), and data integration. Most of the challenges that are described in these sections fall under categories 2 and 3 above (unclear terms and inadequate linking).

There are two additional sections in the Discussion that describe problems facing ontology development: incomplete ontologies and ontology legacy alignment. These ontology-related problems could also be considered to be impediments to progress in relating phenotypes and environments if the authors can successfully make their case that formal ontologies (heavyweight ontologies using OWL to define their terms) are a more effective computational tool than conventional alternatives (category 4 above).

Value of ontologies

The authors state or imply throughout the paper that formal ontologies are the most effective way to solve the problems of standardization and linking, and that their use will lead to more powerful computation through machine reasoning. Here are some statements made in the Introduction that illustrate the authors' view:

61 "…One approach that
62 is likely to improve the status quo involves the use of ontologies to standardize and link data
63 about phenotypes and environments. Specifying and linking data in this manner will allow
64 researchers to increase the scope and flexibility of large-scale analyses aided by modern
65 computing methods."

87 "…An ontology has the potential to tame this heterogeneity and allow researchers to
88 more efficiently query and manipulate, large-scale data sets…"

134 "…Thus, there is a need for a more developed, flexible, and interlinked ontology
135 framework representing environments, phenotypes, and their interplay. This framework for
136 environments and phenotypes can allow automated inferencing over large, aggregated data sets…"

145 "…Thus
146 ontologies empower computers to reliably interpret and reason over these logical relationship
147 graphs. …"

To address the interesting questions about interactions between phenotype and environment that the authors enumerate, they propose that a greater effort be expended in ontology building. In my mind, the value of this paper hinges on whether the authors can make the case that ontologies provide a more effective mechanism for vocabulary standardization, linking, and computing, and therefore are deserving of the proposed effort to build them. The authors present use cases and proof of concept examples to support their case for the power of ontologies.

Use cases

To "communicate the importance of investing in environment and phenotype ontologies", the authors describe use cases that "represent the types of research questions that either cannot currently be answered or can only be answered with great difficulty" (lines 185-187). This led me to believe that the authors would suggest mechanisms by which ontology-based reasoning would make it possible to address the questions.

In each example category (agriculture, taxonomy, phylogeny, behavior), the use cases compare putative current and future workflows to show how current workflows might be enhanced through the use of ontologies. The presentation of pairs of current and future workflows was a bit awkward and required me to scroll back and forth comparing the text of the two workflows to figure out what it was that made the future workflow better than the current workflow. In each case, the future workflow did seem to be "easier" because of some added technology that was not present in the current workflow. To break this down by example:

In the agriculture use case (lines 214-273) the future workflows were improved by using a more tightly controlled vocabulary, a better linked database, and a "semantically aware database" (line 250) or "semantically aware, publicly-available biology database" (lines 267-8). The examples do not explain how the semantics associated with the controlled vocabulary would be used to improve linking or to enable inferencing. It is not clear how the benefits of a "semantically aware" database would manifest themselves.

In the taxonomy use case (lines 313-360), specimen annotation workflow is improved via better software and automation of request for ENVO terms.

In the phylogeny use case (lines 410-444), the improvement in workflow was a visualization tool showing class hierarchy based on mapping to ENVO classes.

In the behavioral use case (lines 486-517), the improvement was that the user "checks a gardening app on his mobile device that uses a combination of ontologies and observation data to power a Q&A engine … Guided by the ontological structure available in its back-end, the app states that … The inference engine used by the app are able to predict …". No information is given about how the inference engine would use the ontologies to make the necessary inferences.

In each of these cases, it was not apparent how the improvements described would have resulted from increased use of ontologies or improved ontologies. In the two examples that imply that the semantics of the ontologies that were used would enable inferencing (the agriculture and behavioral use cases), no information is given about how the ontologies that were used would generate entailments that would provide additional information that would aid in the queries.

Validity of the findings

The authors provide two proof of concept examples that they use to show the value of linking phenotype to environment:

1. Miniaturization in fish. In this example, the authors begin by using the Ontotrace tool to generate a matrix that relates phenotypes to taxa. They use a variety of methods to link the taxa to habitat variables; roughly: Ontotrace taxon names to GBIF (string matching) -> lat/long matching to HydroSHEDS to find watershed -> climate, topography, land cover, geology layers (GIS?) -> habitat variables used in 2-tailed t-tests.

2. Amphibian reproduction. In this example, the authors queried EOL's traitbank to retrieve breeding environment and developmental mode for 282 amphibian taxa. Although they note that EOL uses controlled vocabulary URIs for identification, apparently no semantic information was involved in the analysis - it simply compared the states of the two variables for the species via a chi-squared test.

The statistical analysis of the first example troubles me. The authors say "we tested the hypothesis that miniaturization is correlated with environmental variables", yet they did not conduct a correlation analysis, which would have required that comparing two continuous variables (one of the factors was discontinuous: miniature/non-miniature). Instead they carried out multiple t-tests. There are two reasons why I find this troubling. One is that in a t-test the independent variable is typically the discontinuous one and the dependent variable is typically continuous. Yet in this example, the authors say that their question is whether miniaturization was driven by environmental variables, which would mean that they considered the discontinuous variable (miniaturized/non-miniaturized) to be the dependent variable and the environmental variable (environmental factor) to be the independent variable. So it is not clear to me that a t-test is the right kind of test to use in this circumstance. The second thing that disturbs me is that they are carrying out many separate, unplanned comparisons. Results were shown for mean temperature and mean precipitation (Table 2), but the text implies that many other factors were tested (with unreported results), including factors related to topography, land cover, surface geology, and perhaps additional climate variables. When one carries out many unplanned comparisons, the probability of Type I error increases with the number of comparisons. So the alpha level for each test should be adjusted so that the experiment-wide alpha remains at 0.05 . Without knowing how many tests they carried out, it is impossible to know how that adjustment should have been made. It seems to me that a better analysis to compare a single discontinuous dependent variable with two states to many continuous independent variables would be a single multiple logistic regression. However, I'm still not sure if that would be the best test - if the authors really want to test for correlation, it might be better to use a continuous value to represent the body size (such as mean body mass for the species) and then conduct an actual correlation analysis that compares mean species body mass to each factor. In such an analysis, there is no assumption of cause and effect, which seems the most appropriate in a situation like this.

There is also a problem with the statistical analysis of the second example. The rule of thumb for a valid chi-squared test requires that no more than 20% of the expected counts be less than 5 and that no expected counts be less than 1. I calculated that the expected counts were 0.43, 0.38, 0.88, and 2.3 for paedomorphic associated with freshwater stream, intermittent pond, permanent pond, and terrestrial respectively. In a case like this, the low frequency categories should be combined with other categories (if there is a sensible way to merge categories) or be left out of the analysis.

In addition to these problems with the statistical analysis, I don't really see how these examples serve as a "proof of concept" for the main point of the paper, i.e., that investing in environment and phenotype ontologies will help to answer difficult questions. The authors admit that they "demonstrate how these links can be made with existing tools." (line 688), but I find that very unsatisfying. In the miniature fish case, it would have been much more compelling if they had shown how ontology terms that they would chose to link the various types of resources (taxa to occurrence lat/long to GIS layers to habitat variables) could contribute to some kind of predictive inferencing. In other words - flesh out what it would mean in this circumstance to create a "semantically aware" database that enables "automated inferencing" (terms they used earlier in the paper to describe the value of using terms from formal ontologies). In the amphibian example, it doesn't seem that there is any actual "semantic linking" going on - TraitBank simply has URI values for breeding environment and developmental mode that are downloaded. How would using an ontology contribute to an example like this at all?

Knowledge representation ("basic models")

The section on Knowledge Representation and its associated figures had great potential for helping the authors make their case of the value of ontologies. Figures 4 and 5 presented graphs showing how some pseudodata might be linked using two formal ontologies (OBOE and BCO). The authors state that "OBOE and BCO were developed for different uses cases". What were those use cases? The authors present four categories of use cases earlier in the paper. How could these two models be used to address any of these use cases? What would the merits or downsides be for each of the models in addressing those use cases? What kinds of SPARQL queries would be used to answer questions using each model? What sorts of entailed triples might be materialized during automated inferencing on triples linked using these models? The authors state that "ongoing efforts to align them are likely to lead to a harmonized model that can work for many different use cases" (lines737-8). What are these use cases? In summary, the authors present two models, but do not explain how they are relevant to the rest of the paper.

Overall conclusions about the paper

Even if this paper fell within the scope of PeerJ (i.e., was a Research Article), it is difficult for me to see how this paper could be publishable in its current form. The overall point of the paper is that ontologies are an effective mechanism for answering difficult questions about interactions between phenotypes and environment, yet I failed to find a single example in the paper where the authors actually show how this might be done. One could argue that ontology development in these areas are in the early stages and that it is premature to expect concrete examples. Yet several of the papers cited by this manuscript do actually show how the semantics captured in ontologies can lead to more powerful querying. For example, Henderson et al. (2007) describe how they use inference rules to support OWL reasoning about subsumption or equivalence, and Dececchi et al. (2015) [cited in the paper as Dececchi in press] describe how they used OWL semantics to reason entailed presence and absence in their character matrix. Clearly it is possible to describe, at least in a general way, how the semantics contained in ontologies can make it easier to answer the many questions that the authors raise in the paper.

It might be possible to rewrite this paper as an actual review paper and submit it to a journal that publishes that kind of paper, but the required revisions would probably be substantial since the reviewed material is scattered throughout various sections of the paper. The current structure of the paper isn't really amenable to be transformed into a systematic review of past work on linking phenotypes and environment. I suspect that writing this kind of review is not actually what the authors want to do anyway.

Additional comments

Minor details

Line 442 Typo: "…groundwater -and- [as] an environmental material…"]

Line 1090: Parr C, Wilson N, Schulz K, Leary P, Hammock J, Rice J, Corrigan RJ Jr. is listed as "in press". However, as of 10 August 2015, its status appears to be "minor revision", not "in press": http://www.semantic-web-journal.net/content/traitbank-practical-semantics-organism-attribute-data-0

Lines 1299-1300: I don't understand the citation "Baskauf and Webb unpublished data". Is this indicating that the data used in the example were unpublished data or that unpublished data show that the representation was simplified? Or is this a reference to Fig 3 of http://www.semantic-web-journal.net/content/darwin-sw-darwin-core-based-terms-expressing-biodiversity-data-rdf-0 ?

Line 1303: Typo in Fig 5.A. "identificaiton"

·

Basic reporting

No Comments

Experimental design

It is unclear to me if "Semantic linking of phenotypes and environments: A review" fits within the scope of the journal and this is the primary reason why I have indicated that the manuscript is in need of major revisions. It is both review and case study, which according to the published Scope of the journal, is a better candidate for PeerJPrePrints.

The proof-of-concept analyses that demonstrate the value of linking phenotypes to environments can be considered original primary research, but these need much greater emphasis, methodologies more explicit, and use-cases de-emphasized or eliminated altogether.

Validity of the findings

No Comments

Additional comments

Use Cases

"Future Workflow: Agriculture" : I assume statements here are meant to bolster the argument that ontology-development and -use accelerate Steve's ability to more efficiently and thoroughly identify candidate lines under future climate scenarios. However, the interjection of wild relatives of sorghum into the scenario results in a muddled argument. Wild relatives would need annotations similar/identical to those Steve's company developed for their commercial lines, a very unlikely prospect. Similarly, 245: "Steve’s company has developed and phenotyped a wide range of parental lines that differ in yield response and phenology" assumes that this process itself was not onerous or time consuming and may perhaps have been automated in some undocumented way. If indeed this use-case is meant to illustrate efficiencies (i.e. less time spent gathering raw phenotype data & subsequent analyses) I do not find it particularly convincing. Perhaps a solution would be to illustrate a use-case that identifies candidate lines under multiple, subtly variable climate change scenarios that are unknowable (or better - illogical and subsequently a costly mistake for Steve's company) using the current workflow.

Connecting Specimen Phenotypes to Environment

The use-case as described is disjointed and the point to be made in the Future Workflow is unclear. What I get from the comparison between current and future workflows is that Kate lacks necessary permissions to add to or refine ENVO to suit her needs. This sounds like a socio- rather than a technical problem, whose fix appears not to significantly improve Kate's taxonomic interests. What if Kate's suggested addition to ENVO were rejected, forcing her to later realize her error?

Reconstructing Ancestral Features and Habitats

What I take from this is that use of ENVO may lead to insights that are currently unknowable. If this is the message to be expressed, it could be made more clear.

Including Species Interactions in Habitat Assessments

Very good. I suggest tightening the text by eliminating the duplication in current vs. future workflows.

line 651 - GBIF downloads now include DOIs. In this instance, the DOI is http://doi.org/10.15468/dl.wzpgpe

Reviewer 3 ·

Basic reporting

This is an exciting paper that makes a compelling case for linking of environmental, phenotype and specimen data, with examples and case studies. It is particularly important that the paper is grounded in existing data and ontologies and shows how real research can be conducted in this domain.

Both existing and future uses are clearly demonstrated and a roadmap for the future is presented. The paper is accurate and honest about limitations and difficulties and at the same time hopeful for progress.

The use of existing ontologies and ontology tools is a strong point.

I recommend the publication of the paper and suggest a change in the introduction. The abstract is explicit about the contents and includes a list of examples and assertions. I would like the introduction to make the same point. Tell the reader what to expect and what will be learned. The paper would be stronger with a more direct introductory paragraph.

Experimental design

No comments

Validity of the findings

The examples are clear, unambiguous and supportive of the findings of the paper

Additional comments

This paper will make a significant contribution to the field and is an excellent result for the NSF funded RCN.

---

## Round 0.2 · accepted · Accept

I have gone through your revised draft and response to reviewers and feel you've done a great job responding to the various critiques and concerns. Thank you for paying close attention to those reviews. I feel your paper is now ready for publication and will make a significant impact. Congratulations.